

# Identification of prognostic factors and nomogram model for patients with advanced lung cancer receiving immune checkpoint inhibitors

Xiuqiong Chen[1,2,3,4,*], Zhaona Li[1,2,3,4,*], Jing Zhou[1,2,3,4], Qianhui Wei[1,2,3,4], Xinyue Wang[1,2,3,4] and Richeng Jiang[1,2,3,4]

[1] Tianjin Medical University Cancer Institute and Hospital, National Clinical Research Center for Cancer, Tianjin, China
[2] Key Laboratory of Cancer Prevention and Therapy, Tianjin, China
[3] Tianjin's Clinical Research Center for Cancer, Tianjin, China
[4] Department of Thoracic Oncology, Tianjin Lung Cancer Center, Tianjin Cancer Institute and Hospital, Tianjin Medical University, Tianjin, China
* These authors contributed equally to this work.

Corresponding authors
Xinyue Wang, xwang18@tmu.edu.cn
Richeng Jiang,
jiangricheng@tjmuch.com

## ABSTRACT

**Background and aim:** Some patients with lung cancer can benefit from immunotherapy, but the biomarkers that predict immunotherapy response were not well defined. Baseline characteristic of patients may be the most convenient and effective markers. Therefore, our study was designed to explore the association between baseline characteristics of patients with lung cancer and the efficacy of immunotherapy.

**Methods:** A total of 216 lung cancer patients from Tianjin Medical University Cancer Institute & Hospital who received immunotherapy between 2017 and 2021 were included in the retrospective analysis. All baseline characteristic data were collected and then univariate log-rank analysis and multivariate COX regression analysis were performed. Kaplan–Meier analysis was used to evaluate patients' progression-free survival (PFS). A nomogram based on significant biomarkers was constructed to predict PFS rate of patients receiving immunotherapy. We evaluated the prediction accuracy of nomogram using C-indices and calibration curves.

**Results:** Univariate analysis of all collected baseline factors showed that age, clinical stage, white blood cell (WBC), lymphocyte (LYM), monocyte (MON), eosinophils (AEC), hemoglobin (HB), lactate dehydrogenase (LDH), albumin (ALB) and treatment line were significantly associated with PFS after immunotherapy. Then these 10 risk factors were included in a multivariate regression analysis, which indicated that age (HR: 1.95, 95% CI [1.01–3.78], $P = 0.048$), MON (HR: 1.74, 95% CI [1.07–2.81], $P = 0.025$), LDH (HR: 0.59, 95% CI [0.36–0.95], $P = 0.030$), and line (HR: 0.57, 95% CI [0.35–0.94], $P = 0.026$) were significantly associated with PFS in patients with lung cancer receiving immunotherapy. Patients with higher ALB showed a greater trend of benefit compared with patients with lower ALB (HR: 1.58, 95% CI [0.94–2.66], $P = 0.084$). Patients aged ≥51 years, with high ALB, low LDH, first-line immunotherapy, and high MON had better response rates and clinical benefits. The nomogram based on age, ALB, MON, LDH, line was established to predict the prognosis of patients treated with immune checkpoint inhibitor (ICI).

The C-index of training cohort and validation cohort were close, 0.71 and 0.75, respectively. The fitting degree of calibration curve was high, which confirmed the high prediction value of our nomogram.

**Conclusion:** Age, ALB, MON, LDH, line can be used as reliable predictive biomarkers for PFS, response rate and cancer control in patients with lung cancer receiving immunotherapy. The nomogram based on age, ALB, MON, LDH, line was of great significance for predicting 1-year-PFS, 2-year-PFS and 3-year-PFS in patients with advanced lung cancer treated with immunotherapy.

## INTRODUCTION

In recent years, immunotherapy drugs targeting immune checkpoints have developed rapidly and made great breakthroughs in tumor therapy. A large number of clinical trials have demonstrated that immunotherapy can bring greater clinical benefit to patients with PD-L1 ≥50% and has rapidly become the standard management of advanced lung cancer (*Ettinger et al., 2021*; *Herbst et al., 2020*; *Reck et al., 2016*; *Mok et al., 2019*). However, only about 30% of patients with advanced lung cancer have PD-L1 levels ≥50% (*Paz-Ares et al., 2018*). Most patients receiving ICI did not experience clinical benefit. Some patients with low PD-L1 expression level can also respond to immunotherapy, while patients with high PD-L1 expression level may not respond to immunotherapy. Therefore, PD-L1 was not a perfect predictor of immunotherapy efficacy (*Borghaei et al., 2020*). Tumor mutation burden (TMB) has also been considered as a predictive biomarker of immunotherapy. The higher the TMB was, the higher the exposure level of neo-antigen *in vivo* was, the more the anti-tumor immune response can be activated (*Sesma, Pardo & Cruellas, 2020*). The Checkmate-227 study indicated that TMB may help clinicians decide whether to use immunotherapy (*Hellmann, Ciuleanu & Pluzanski, 2018*). However, further updated data from this research showed that the extension of OS in lung cancer patients treated with Nivolumab and Ipilimumab was not associated with TMB and PD-L1 (*Hellmann, Paz-Ares & Caro, 2019*). Therefore, the predictive role of these markers were still controversial. PD-L1 and TMB were commonly detected using tumor tissue, which was difficult to obtain and expensive. There was an urgent need to explore a reliable and available biomarker to help clinicians identify patients whose tumors were more likely to respond to immunotherapy agents and whose survival was more likely to be prolonged. That may help enhance the precision of treatment and improve the cost-effectiveness with fewer toxicity associated with treatments.

Peripheral hematological biomarkers were the main direction of our exploration. As we all know, peripheral blood indicators including white blood cells, neutrophils, lymphocytes, platelets, albumin, lactate dehydrogenase, C-reactive protein and others can reflect the systemic inflammatory of the body. After immunotherapy, tumor cells had more antigen exposure or the killing function of effective leukocyte were activated, MON was polarized towards M1 macrophages, eosinophils secreted cytokines to regulate the function of T cells,

and the proportion of Treg cells was reduced (*Chu et al., 2020*; *Petitprez et al., 2020*). These changes played a role in clearing tumor cells from circulating blood. Literatures has reported that the tumor microenvironment was in a state of hypoxia, high LDH increased glycolysis and lactic acid production from tumor cells and immune cells in the microenvironment. Therefore, the amount of LDH may reflect the immune state of the body and determine whether it was suitable for immunotherapy (*Kumagai et al., 2022*). HB and ALB were not only transport mediators in the body, but also nutritional markers in the body, which were crucial for the use of immunotherapy. Inflammation was a major driver of cancer development, which may represented the balance between anti-tumor immune response and pro-inflammatory state (*Diakos et al., 2014*). Changes in these inflammatory mediators can lead to clinical physiological symptoms such as pain. Pain was a major symptom of cancer and has been identified to be related to systemic inflammation. Connections between other cancer symptoms and systemic inflammation have also been explored, suggesting that systemic inflammation may be involved in the development and progression of cancer (*Laird et al., 2013*). Therefore, the state of systemic inflammation can reflect the state of tumors in the body and partly reflect the immune state of the body. It was better to evaluate the immune state of the body before immunotherapy to help the clinician to make a decision.

Numerous researches have described some hematologic indicators such as NLR, MLR and PLR as independent prognostic factors for various cancer types (*Zhang et al., 2021*; *Song et al., 2021*; *Jin et al., 2021*; *Cupp et al., 2020*; *Ethier et al., 2017*; *Liu et al., 2021*; *Gao et al., 2020*). However, the association between these biomarkers and PFS of patients with lung cancer receiving immunotherapy or response of tumor to immunotherapy was still unknown. Many studies on hematologic biomarkers have focused on the neutrophil to lymphocyte ratio (NLR) (*Chu et al., 2020*; *Valero et al., 2021*; *Capone et al., 2018*; *Bartlett et al., 2020*; *Hata et al., 2020*), but the results were controversial. Some studies had only investigated the correlation between peripheral blood lymphocytes or their subsets and the prognosis of immunotherapy, but other peripheral blood indicators have not been further explored (*Xu et al., 2021*). The investigation between MLR, PLR or other hematological indicators and the efficacy of immunotherapy was still lacking. Therefore, our study comprehensively analyzed the systemic hematologic markers of lung cancer patients, in an attempt to excavate reliable biomarkers conducive to the evaluation of immune status, which can be used to assist in the selection of clinical treatments.

## MATERIALS AND METHODS

Patient selection: A total of 216 patients with pathologically diagnosed as lung cancer including adenocarcinoma, squamous cell carcinoma and others from the Tianjin Medical University Cancer Institute & Hospital were included in our study. All patients were treated with combined or mono-immunotherapy for at least two cycles. Assessment was performed every two cycles. Prior to initiation of immunotherapy, baseline hematologic indicators or clinical characteristics of patients were recorded and can be collected. Clinical response assessment was performed by documented radiographic reports of complete response (CR), partial response (PR), stable disease (SD), or progressive disease (PD)

according to response evaluation criteria in solid tumors (RECIST 1.1 version) (*Wolchok et al., 2009*). Treatment was discontinued when the patient progressed or developed an unacceptable immune-related adverse reaction. Patients with multiple concurrent tumors, less than two cycles of immunotherapy, and no hematology tests performed within 30 days prior to immunotherapy were excluded.

Patient parameter: Baseline assessments were defined as those taken within 2 weeks before receiving immunotherapy. These baseline characteristics included gender, age, pathological features, smoking status, body surface area (BSA), BMI, clinical stage, treatment line, recurrence after radical surgery, previous thoracic radiation therapy, and combination or monotherapy. The baseline peripheral blood biomarkers include white blood cells, neutrophils, lymphocytes, platelets, eosinophils, monocytes, hemoglobin, lactate dehydrogenase, albumin, alkaline phosphatase, and C-reactive protein.

Statistical methods: Progression-free survival time (PFS) was defined as the time from initiation of a treatment regimen to disease progression or death. Efficacy was assessed according to RECIST 1.1 criteria, and disease progression was defined as an increase of at least ≥20% in the sum of the maximum diameters of the target lesions, or the emergence of new lesions. The objective response rate (ORR) or response rate was considered as the proportion of patients with partial or complete response for at least 1 month after evaluation based on RESIST criteria. Disease control rate (DCR) or clinical benefits was the total proportion of patients with complete remission, partial remission and stable disease. Statistical software X-tile was used to intercept the optimal cut-off values for continuous variables including age, BSA, BMI and peripheral blood biomarkers. IBM-SPSS version 26.0 was applied to make survival curve and establish COX proportional regression model. $P$ values less than 0.05 were considered statistically significant. Univariate analysis was performed on all factors that might affect the efficacy of immunotherapy, and risk factors with $P$ value less than 0.05 were included in multivariate analysis. The Kaplan-Meier method was employed to draw survival curves between different groups, and log-rank test was applied to compare the statistical differences between the two groups. We applied R software (version 3.6.1; *R Core Team, 2019*) to establish nomogram based on independent prognostic factors and internal validation was used to verify the reliability of nomogram.

## RESULTS

### Clinical features of patients with lung cancer receiving immunotherapy

The baseline characteristics of 216 patients enrolled in our study were shown in Table 1. Male smokers accounted for the majority of patients, about 80.6% and 70.4%. A total of 84.7% of the patients were older than 51 years. Of the 216 patients, 80 received first-line immunotherapy, 129 received subsequent line immunotherapy, and the remaining seven patients had unknown treatment line. The most common type of pathology was lung adenocarcinoma. 36.1% of the patients had a body surface area of less than 1.94, and 30.1% had a BMI of less than 28.1. A total of 75.9% of patients were at stage IV. Baseline peripheral hematologic markers including WBC, NEU, LYM, PLT, MON, AEC, HB, CRP, LDH, ALP, and ALB were included in the study. NLR and LMR were the ratio of neutrophils to lymphocytes and the ratio of lymphocytes to monocytes, respectively. PLR

**Table 1  Baseline patient characteristics.**

| Clinical parameters | | Total number (%) | Training cohort (%) | Validation cohort (%) |
|---|---|---|---|---|
| Gender | Male | 174 (80.6%) | 73 (76.04%) | 33 (89.19%) |
| | Female | 42 (19.4%) | 23 (23.96%) | 4 (10.81%) |
| Age | <51 years old | 33 (15.3%) | 15 (15.63%) | 2 (5.41%) |
| | ≥51 years old | 183 (84.7%) | 81 (84.38%) | 35 (94.59%) |
| Pathology | Adenocarcinoma | 99 (45.8%) | 43 (44.79%) | 18 (48.65%) |
| | SCC | 78 (36.1%) | 35 (36.46%) | 12 (32.43%) |
| | Others | 33 (15.3%) | 17 (17.71%) | 6 (16.22%) |
| | Unknown | 6 (2.8%) | 1 (1.04%) | 1 (2.70%) |
| Smoking | Yes | 152 (70.4%) | 63 (65.63%) | 28 (75.68%) |
| | No | 47 (21.8%) | 25 (26.04%) | 7 (18.92%) |
| | Unknown | 17 (7.9%) | 8 (8.33%) | 2 (5.41%) |
| BSA | <1.94 | 78 (36.1%) | 42 (43.75%) | 18 (48.65%) |
| | ≥1.94 | 19 (8.8%) | 7 (7.29%) | 5 (13.51%) |
| | Unknown | 119 (55.1%) | 47 (48.96%) | 14 (37.84%) |
| BMI | <28.1 | 65 (30.1%) | 33 (34.38%) | 16 (43.24%) |
| | ≥28.1 | 8 (3.7%) | 4 (4.17%) | 1 (2.70%) |
| | Unknown | 143 (66.2%) | 59 (61.46%) | 20 (54.05%) |
| Clinical stage | III | 42 (19.4%) | 19 (19.79%) | 3 (8.11%) |
| | IV | 164 (75.9%) | 73 (76.04%) | 34 (91.89%) |
| | Unknown | 10 (4.6%) | 4 (4.17%) | 0 (0.00%) |
| WBC | <6.59 | 106 (49.1%) | 52 (54.17%) | 17 (45.95%) |
| | ≥6.59 | 95 (44%) | 44 (45.83%) | 20 (54.05%) |
| | Unknown | 15 (6.9%) | 0 (0.00%) | 0 (0.00%) |
| NEU | <4.80 | 128 (59.3%) | 66 (68.75%) | 21 (56.76%) |
| | ≥4.80 | 73 (33.8%) | 30 (31.25%) | 16 (43.24%) |
| | Unknown | 15 (6.9%) | 0 (0.00%) | 0 (0.00%) |
| LYM | <0.86 | 34 (15.8%) | 13 (13.54%) | 5 (13.51%) |
| | ≥0.86 | 167 (77.3%) | 83 (86.46%) | 32 (86.49%) |
| | Unknown | 15 (6.9%) | 0 (0.00%) | 0 (0.00%) |
| PLT | <323 | 173 (80.1%) | 80 (83.33%) | 32 (86.49%) |
| | ≥323 | 28 (13%) | 16 (16.67%) | 5 (13.51%) |
| | Unknown | 15 (6.9%) | 0 (0.00%) | 0 (0.00%) |
| MON | <0.50 | 75 (34.7%) | 33 (34.38%) | 14 (37.84%) |
| | ≥0.50 | 126 (58.4%) | 63 (65.63%) | 23 (62.16%) |
| | Unknown | 15 (6.9%) | 0 (0.00%) | 0 (0.00%) |
| AEC | <0.24 | 157 (72.7%) | 74 (77.08%) | 25 (67.57%) |
| | ≥0.24 | 44 (20.4%) | 22 (22.92%) | 12 (32.43%) |
| | Unknown | 15 (6.9%) | 0 (0.00%) | 0 (0.00%) |
| HB | <130 | 105 (48.6%) | 53 (55.21%) | 15 (40.54%) |
| | ≥130 | 96 (44.5%) | 43 (44.79%) | 22 (59.46%) |
| | Unknown | 15 (6.9%) | 0 (0.00%) | 0 (0.00%) |

(Continued)
| Table 1 (continued) | | | | |
|---|---|---|---|---|
| Clinical parameters | | Total number (%) | Training cohort (%) | Validation cohort (%) |
| CRP | <32.09 | 33 (15.3%) | 12 (12.50%) | 7 (18.92%) |
| | ≥32.09 | 10 (4.6%) | 4 (4.17%) | 3 (8.11%) |
| | Unknown | 173 (80.1%) | 80 (83.33%) | 27 (72.97%) |
| LDH | <241 | 77 (35.7) | 53 (55.21%) | 20 (54.05%) |
| | ≥241 | 61 (28.2%) | 43 (44.79%) | 17 (45.95%) |
| | Unknown | 78 (36.1%) | 0 (0.00%) | 0 (0.00%) |
| ALP | <109 | 144 (66.7%) | 69 (71.88%) | 26 (70.27%) |
| | ≥109 | 56 (25.9%) | 27 (28.13%) | 11 (29.73%) |
| | Unknown | 16 (7.4%) | 0 (0.00%) | 0 (0.00%) |
| ALB | <37.8 | 55 (25.5%) | 26 (27.08%) | 10 (27.03%) |
| | ≥37.8 | 146 (67.6%) | 70 (72.92%) | 27 (72.97%) |
| | Unknown | 15 (6.9%) | 0 (0.00%) | 0 (0.00%) |
| PLR | <231.8 | 138 (63.9%) | 66 (68.75%) | 30 (81.08%) |
| | ≥231.8 | 63 (29.2%) | 30 (31.25%) | 7 (18.92%) |
| | Unknown | 15 (6.9%) | 0 (0.00%) | 0 (0.00%) |
| NLR | <3.67 | 123 (57%) | 60 (62.5%) | 27 (72.97%) |
| | ≥3.67 | 78 (36.1%) | 36 (37.50%) | 10 (27.03%) |
| | Unknown | 15 (6.9%) | 0 (0.00%) | 0 (0.00%) |
| LMR | <4.29 | 169 (78.3%) | 86 (89.58%) | 29 (78.38%) |
| | ≥4.29 | 32 (14.8%) | 10 (10.42%) | 8 (21.62%) |
| | Unknown | 15 (6.9%) | 0 (0.00%) | 0 (0.00%) |
| Line | First line | 80 (37%) | 35 (36.46%) | 12 (32.43%) |
| | After first line | 129 (59.7%) | 61 (63.54%) | 25 (67.57%) |
| | Unknown | 7 (3.3%) | 0 (0.00%) | 0 (0.00%) |
| Chest radiotherapy | Yes | 50 (23.1%) | 21 (21.88%) | 9 (24.32%) |
| | No | 164 (75.9%) | 74 (77.08%) | 28 (75.68%) |
| | Unknown | 2 (1%) | 1 (1.04%) | 0 (0.00%) |
| Radical surgery | Yes | 55 (25.5%) | 78 (81.25%) | 9 (24.32%) |
| | No | 161 (74.5%) | 18 (18.75%) | 28 (75.68%) |
| | <50% | 14 (6.48%) | 6 (6.25%) | 3 (8.11%) |
| PD-L1 | ≥50% | 20 (9.26%) | 12 (12.5%) | 2 (5.40%) |
| | Unknown | 182 (84.26%) | 78 (81.25%) | 32 (86.49%) |
| Drug use | ICI alone | 71 (32.9%) | 38 (39.58%) | 12 (32.43%) |
| | ICI combination | 145 (67.1%) | 58 (60.42%) | 25 (67.57%) |
| Types of ICIs | Pembrolizumab | 150 (69.44%) | 61 (63.54%) | 30 (81.08%) |
| | Tislelizumab | 38 (17.59%) | 21 (21.88%) | 5 (13.51%) |
| | Others | 28 (12.97%) | 14 (14.58%) | 2 (5.41%) |

was defined as the ratio of platelets to lymphocytes. X-tile were used to calculate the optimal thresholds for continuity variables, and patients were then divided into two groups based on the thresholds. Table 1 showed the optimal cut-offs for all variables and the proportion of patients in the two groups.

## Univariate and multivariate analyses of biomarkers for PFS

The median PFS for all patients was 11.4 months, and the results of univariate and multifactorial analyses were listed in Table 2. After univariate analysis, risk factors with $P < 0.05$ were included in multivariate analysis. Finally, age (HR: 1.94, 95% CI [1.22–3.09], $P = 0.005$), clinical stage (HR: 0.58, 95% CI [0.34–0.99], $P = 0.049$), WBC (HR: 1.58, 95% CI [1.07–2.32], $P = 0.021$), LYM (HR: 1.79, 95% CI [1.12–2.86], $P = 0.014$), AEC (HR: 2.17, 95% CI [1.26–3.76], $P = 0.005$), MON (HR: 1.58, 95% CI [1.08–2.33], $P = 0.020$), HB (HR: 1.71, 95% CI [1.16–2.52], $P = 0.006$), LDH (HR: 0.63, 95% CI [0.40–0.98], $P = 0.043$), ALB (HR: 1.79, 95% CI [1.19–2.69], $P = 0.005$), and Line (HR: 0.63, 95% CI [0.42–0.95], $P = 0.029$) were included in COX regression analysis. Multivariate analysis indicated that age, LDH, MON, ALB and line were significantly correlated with PFS. Patients aged ≥51 years who received immunotherapy in the first line were more likely to benefit from immunotherapy than those aged <51 years who received immunotherapy in the back line (Figs. 1A, 1E). Patients with LDH <241 had significantly longer PFS than those with LDH ≥241 (Fig. 1C), with a median PFS of 15.5 months in the LDH <241 group and 8.67 months in the LDH ≥241 group, indicating a statistically significant difference between the two groups (HR: 0.59, 95% CI [0.36–0.95], $P = 0.030$). MON was also an important independent prognostic factor, and patients with MON ≥0.5 have a higher probability to benefit from ICI (HR: 1.74, 95% CI [1.07–2.81], $P = 0.025$, Fig. 1B). Patients with ALB ≥37.8 showed a greater trend of benefit compared with patients with ALB <37.8 (HR: 1.58, 95% CI [0.94–2.66], $P = 0.084$, Fig. 1D).

## Response rate and clinical benefit rate between different groups

In patients aged <51 years group, ORR and DCR were 30.30% and 72.73%, respectively, which were lower than those in age ≥51 years group (32.14% and 76.79%, respectively). Patients with lung cancer receiving ICI in the first line had better response rates (47.22% and 21.71%) and clinical benefits (81.94% and 70.54%) than those who using ICI after first line. In terms of blood biomarkers, the DCR and ORR of LDH <241 group (76.62% and 35.06%) were significantly higher than those of LDH ≥241 group (66.13% and 25.81%). Considering the ALB, the difference of clinical benefits (78.08% and 56.36%) between two groups was larger than response rates (29.09% and 30.14%). Patients with higher MON also had better DCR (73.02% and 69.33%) and ORR (32.54% and 26.67%) than those with lower MON. The results were displayed in Table 3.

## Building and validating the nomogram for PFS

After multivariate regression, significant indicators including age, Line, ALB, LDH, and MON were used to develop the predictive nomogram. Patients with incomplete data related to age, Line, ALB, LDH, and MON were excluded, and the remaining 133 patients

**Table 2 Univariate and multivariate analyses of factors associated with PFS of lung cancer patients treated with immunotherapy.**

| | Univariate analysis | | Multivariate analysis | |
|---|---|---|---|---|
| | HR (95% CI) | P-value | HR (95% CI) | P-value |
| Age (years) | | 0.005 | | 0.048 |
| <51 | 1.94 [1.22–.09] | | 1.95 [1.01–3.78] | |
| ≥51 | Ref | | Ref | |
| Gender | | 0.793 | | |
| Female | 1.07 [0.67–1.70] | | | |
| Male | Ref | | | |
| Pathology | | 0.938 | | |
| Adenocarcinoma | 1.08 [0.63–1.85] | | | |
| SCC | 1.11 [0.64–1.94] | | | |
| Others | Ref | | | |
| Smoking | | 0.435 | | |
| Yes | 0.84 [0.54–1.31] | | | |
| No | Ref | | | |
| BSA | | 0.063 | | |
| <1.94 | 1.96 [0.96–4.00] | | | |
| ≥1.94 | Ref | | | |
| BMI | | 0.118 | | |
| <28.1 | 0.50 [0.21–1.19] | | | |
| ≥28.1 | Ref | | | |
| Clinical stage | | 0.049 | | |
| III | 0.58 [0.34–0.99] | | | |
| IV | Ref | | | |
| WBC | | 0.021 | | |
| <6.59 | 1.58 [1.07–2.32] | | | |
| ≥6.59 | Ref | | | |
| NEU | | 0.149 | | |
| <4.80 | 1.35 [0.90–2.04] | | | |
| ≥4.80 | Ref | | | |
| LYM | | 0.014 | | |
| <0.86 | 1.79 [1.12–2.86] | | | |
| ≥0.86 | Ref | | | |
| PLT | | 0.067 | | |
| <323 | 1.90 [0.96–3.76] | | | |
| ≥323 | Ref | | | |
| MON | | 0.020 | | 0.025 |
| <0.50 | 1.58 [1.08–2.33] | | 1.74 [1.07–2.81] | |
| ≥0.50 | Ref | | Ref | |
| AEC | | 0.005 | | |
| <0.24 | 2.17 [1.26–3.76] | | | |

| Table 2 (continued) | | | | |
|---|---|---|---|---|
| | Univariate analysis | | Multivariate analysis | |
| | HR (95% CI) | *P*-value | HR (95% CI) | *P*-value |
| ≥0.24 | Ref | | | |
| HB | | 0.006 | | |
| <130 | 1.71 [1.16–2.52] | | | |
| ≥130 | Ref | | | |
| CRP | | 0.074 | | |
| <32.09 | 0.46 [0.20–1.08] | | | |
| ≥32.09 | Ref | | | |
| LDH | | 0.043 | | 0.030 |
| <241 | 0.63 [0.40–0.98] | | 0.59 [0.36–0.95] | |
| ≥241 | Ref | | Ref | |
| ALP | | 0.055 | | |
| <109 | 1.54 [0.99–2.39] | | | |
| ≥109 | Ref | | | |
| ALB | | 0.005 | | 0.084 |
| <37.8 | 1.79 [1.19–2.69] | | 1.58 (0.94-2.66) | |
| ≥37.8 | Ref | | Ref | |
| PLR | | 0.074 | | |
| <231.8 | 0.70 [0.47–1.04] | | | |
| ≥231.8 | Ref | | | |
| NLR | | 0.204 | | |
| <3.67 | 0.78 [0.53–1.15] | | | |
| ≥3.67 | Ref | | | |
| LMR | | 0.138 | | |
| <4.29 | 0.68 [0.41–1.13] | | | |
| ≥4.29 | Ref | | | |
| Line | | 0.029 | | 0.026 |
| First line | 0.63 [0.42–0.95] | | 0.57 [0.35–0.94] | |
| After first line | Ref | | Ref | |
| Chest radiotherapy | | 0.555 | | |
| Yes | 1.14 [0.75–1.73] | | | |
| No | Ref | | | |
| Radical surgery | | 0.479 | | |
| Yes | 0.86 [0.55–1.32] | | | |
| No | Ref | | | |
| PD-L1 | | 0.503 | | |
| <50% | 1.35 [0.56–3.22] | | | |
| ≥50% | Ref | | | |
| Drug use | | 0.859 | | |
| ICI alone | 1.04 [0.70–1.53] | | | |
| ICI combination | Ref | | | |

(Continued)

| | Univariate analysis | | Multivariate analysis | |
|---|---|---|---|---|
| | HR (95% CI) | *P*-value | HR (95% CI) | *P*-value |
| Types of ICIs | | 0.643 | | |
| Pembrolizumab | 0.83 [0.49–1.42] | | | |
| Tislelizumab | 0.74 [0.39–1.39] | | | |
| Others | Ref | | | |

were used to create and validate the nomogram. These available patients were randomly assigned to the training cohort and testing cohort at a 7:3 ratio. The training cohort was applied to establish the predictive nomogram and internal verification was performed in the testing cohort. The nomogram predicting the 1-, 2-, and 3-year PFS of patients was virtually presented in Fig. 2. The C index of the training cohort was 0.71, while this value was 0.75 in the validation cohort. In addition, the calibration curves showed a great consistency between the actual probabilities of 1-, 2-, and 3-year PFS and the predicted results in both the training and testing cohorts (Fig. 3). These results manifested the good performance and application of our prediction model.

## DISCUSSION

Earlier researches have shown that chronic inflammation predisposed people to various types of cancer, and that inflammatory responses were associated with 15–20% of cancer-related deaths (*Mantovani et al., 2008*; *Balkwill & Mantovani, 2001*). Many studies have indicated that inflammation can aid malignant tumor cells survive and promote tumor development and metastasis (*Hu, Ma & Hu, 2018*; *Karki, Man & Kanneganti, 2017*; *Grivennikov, Greten & Karin, 2010*). Therefore, the inflammatory status of the organism was closely related to the strength of immune response. Immune checkpoint inhibitors such as PD-L1, PD-1 and CTLA-4 antibodies all assist the body's own immune cells to exert cytotoxic effects to kill cancer cells. However, studies have proved that the expression level of PD-L1 on tumor cell surface still has limitations as a criterion for patients to choose immunotherapy and to predict the prognosis of immunotherapy patients (*Alexander, McMillan & Park, 2021*; *Martin & Märkl, 2019*; *Davis & Patel, 2019*). An analysis of all US Food and Drug Administration (FDA) approvals of 45 immune checkpoint inhibitors across 15 tumor types found that PD-L1 was predictive in only 28.9% of cases, and was either not predictive (53.3%) or not tested (17.8%) in the remaining cases (*Davis & Patel, 2019*). This research reported that the PD-L1 thresholds were variable within or across cancer types using several different assays and PD-L1 expression was also evaluated in a variable fashion either on tumor cells, tumor-infiltrating immune cells, or both, which indicated that PD-L1 expression as a predictive biomarker had shortcomings and that the decision to pursue testing must be carefully implemented for clinical decision-making. Therefore, it was of great significance to explore efficient and reliable biomarkers to predict the efficacy of immunotherapy. In addition to immune cells in the tumor

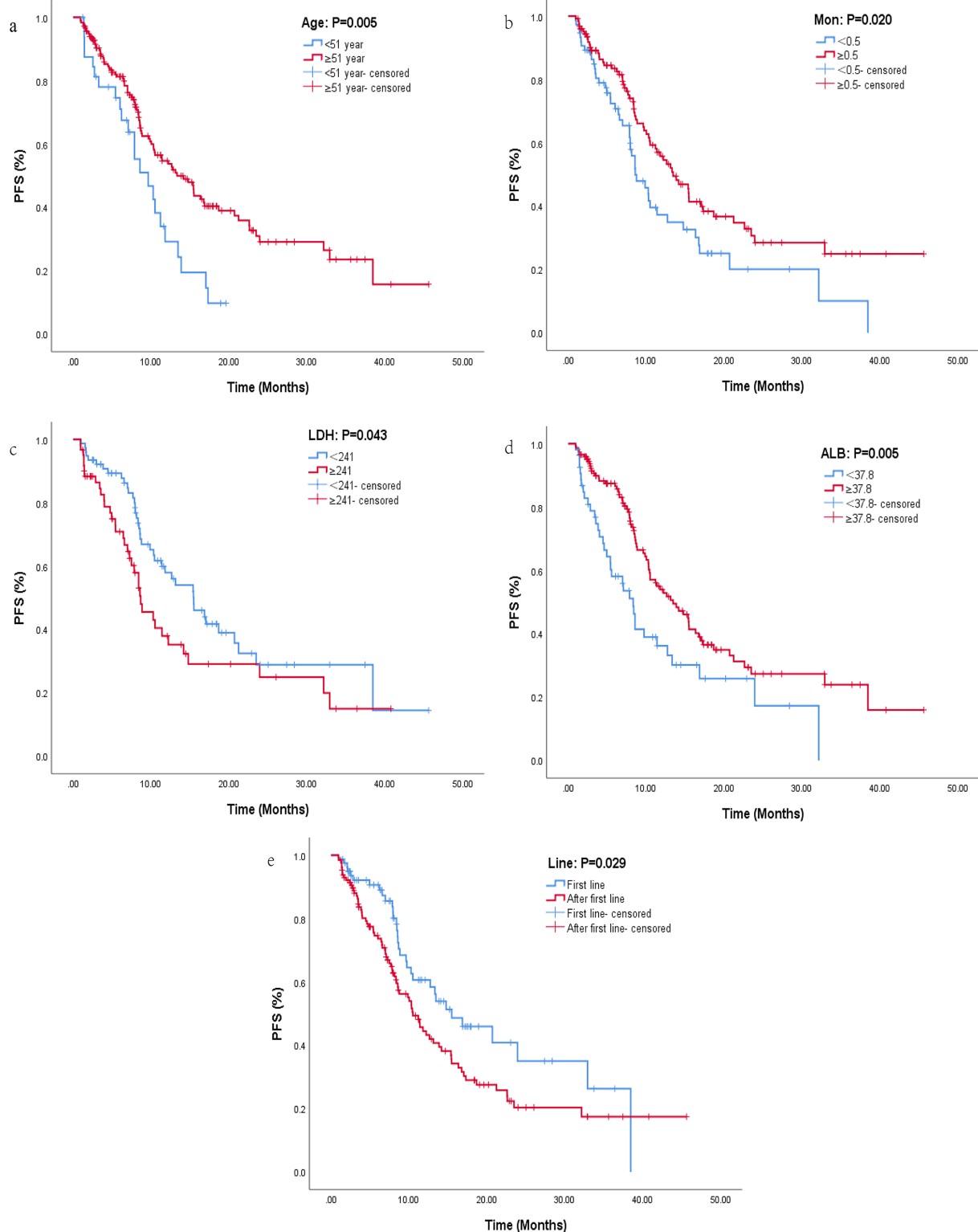

**Figure 1 Kaplan–Maier curves of progression-free survival for lung cancer patients receiving immunotherapy in different groups.** Kaplan–Maier curves of progression-free survival for lung cancer patients receiving immunotherapy in different groups. (A) Kaplan–Maier curve for different ages. (B) Kaplan–Maier curve for different levels of MON. (C) Kaplan–Maier curve for different levels of LDH. (D) Kaplan–Maier curve for different levels of ALB. (E) Kaplan–Maier curve for different lines of immunotherapy.

**Table 3 Response and clinical benefits of patients in different groups and combinations.**

| Group | | Response rate | Clinical benefits |
|---|---|---|---|
| Age | <51 years | 30.30% | 72.73% |
| | ≥51 years | 32.14% | 76.79% |
| Mon | <0.5 | 26.67% | 69.33% |
| | ≥0.5 | 32.54% | 73.02% |
| Line | First line | 47.22% | 81.94% |
| | After first line | 21.71% | 70.54% |
| ALB | <37.8 | 29.09% | 56.36% |
| | ≥37.8 | 30.14% | 78.08% |
| LDH | <241 | 35.06% | 76.62% |
| | ≥241 | 25.81% | 66.13% |

microenvironment, peripheral leucocytes were also targets of immune checkpoints inhibitors. After immunotherapy, the changes of hematological indicators including WBC, LYM, MON and AEC directly affected the efficacy of immunotherapy. Therefore, baseline peripheral blood indicators may be predictors of immunotherapy efficacy. Several studies have shown that inflammatory biomarkers and nutritional indicators such as NLR, Prognostic nutrition Index (PNI), and LDH were associated with prognosis and immunotherapy outcomes in cancer patients. The lower NLR, the lower LDH, and the higher PNI, the better the outcome of immunotherapy (*Peng et al., 2020*; *Ho et al., 2018*). At present, NLR was the most studied. A research examining the association of baseline NLR with the efficacy of whole-cancer immunotherapy indicated that lower NLR was significantly associated with better OS and PFS, and lower rates of response and clinical benefit after ICI therapy across multiple cancer types, When NLR combined with tumor mutation load (TMB) predicted the efficacy of immunotherapy, patients in the low NLR/high TMB group were significantly more likely to benefit from immunotherapy than those in the high NLR/Low TMB group (OR = 3.22, 95% CI [2.26–4.58], $P = 0.001$) (*Valero et al., 2021*). This study established the predictive value of neutrophils and lymphocytes in cancer immunotherapy, especially for non-small cell lung cancer. Sixteen different cancers were included in the study, with small sample sizes for all but non-small cell lung cancer. Another study exploring the correlation between peripheral blood biomarkers and the efficacy of immunotherapy in non-small cell lung cancer showed no significant association between baseline neutrophils and lymphocytes with PFS and OS (*Chu et al., 2020*). Our study took this controversial conclusion a step further by comprehensively analyzing every possible indicator of immunotherapy efficacy.

Results showed that age, line, ALB, LDH, and MON can be applied as one of the criteria for patients of lung cancer to choose immunotherapy. Older patients with higher MON, lower LDH and higher ALB had better PFS, response rates and clinical benefit with first-line immunotherapy. ALB and LDH, as indicators of nutritional status and inflammatory status respectively, had been reported to be significantly correlated with the prognosis of a variety of tumors (*Shen, Wang & Yu, 2021*; *Xie et al., 2022*). In the tumor

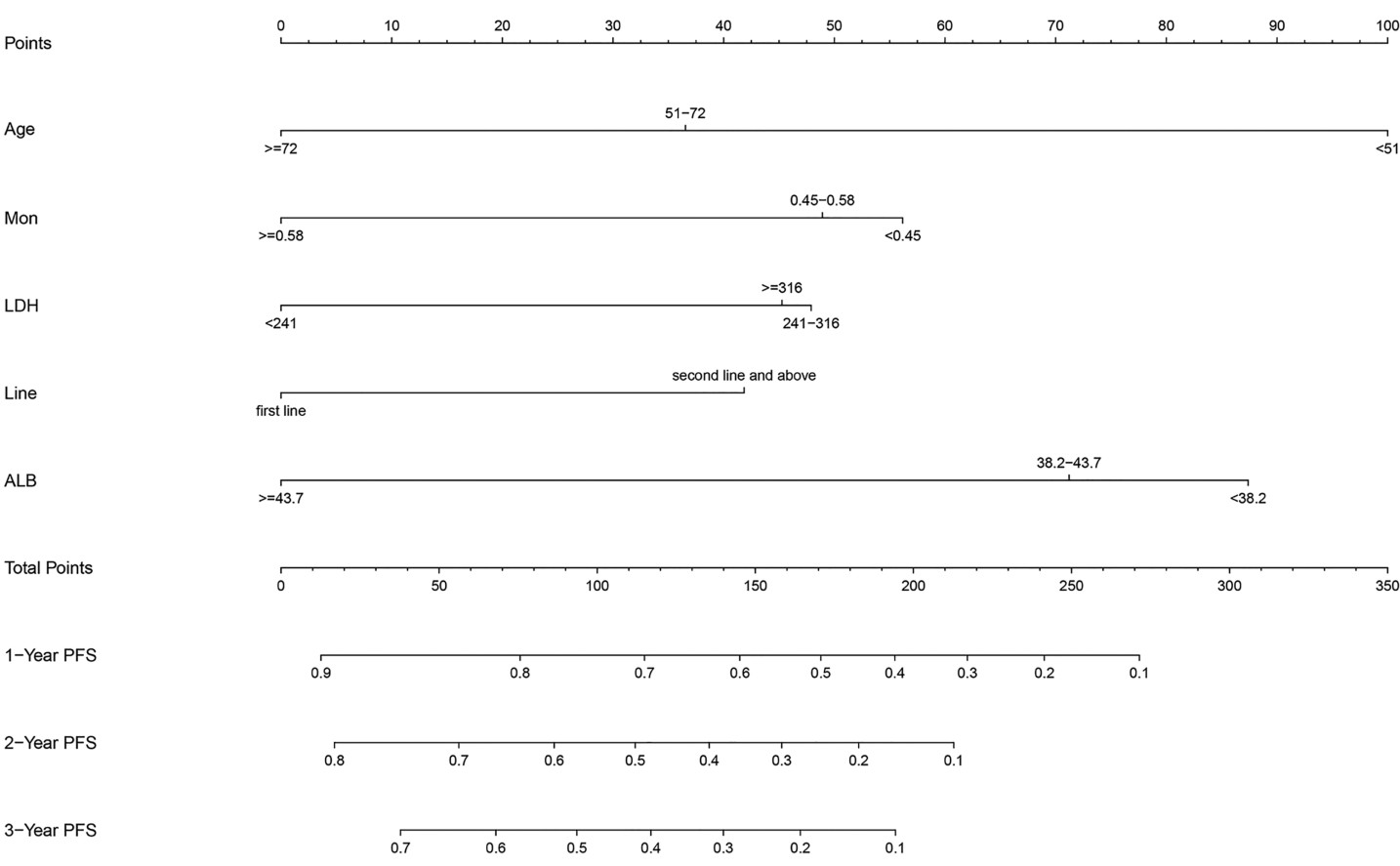

**Figure 2** The nomogram containing identified factors for the 1-, 2-, and 3-year PFS prediction of lung cancer patients receiving ICI.

microenvironment, with abnormal proliferation of tumors, lactic acid was produced in large quantities, and LDH was a key enzyme that was required for the conversion of pyruvate to lactic acid. A large amount of lactic acid accumulation in the microenvironment can lead to the differentiation of Treg cells, resulting in a suppressed microenvironment (*Kumagai et al., 2022*). Several clinical studies have reported that LDH can predict the efficacy of immunotherapy in melanoma patients (*Kelderman et al., 2014*; *Diem et al., 2016*). For lung cancer patients, studies have also shown that LDH can be used to predict PFS (*Taniguchi et al., 2017*) and OS (*Mezquita et al., 2018*) after immunotherapy. Our study further confirmed the predictive value of LDH in immunotherapy for lung cancer patients.

Many studies have suggested that nutritional status was an important predictor of tumor prognosis and albumin level was the main embodiment of body nutrition (*Shen, Wang & Yu, 2021*; *Yu et al., 2021*; *Fang et al., 2021*; *Suh et al., 2014*). Our study was the first to demonstrate that albumin can be applied to predict the efficacy of immunotherapy in patients with lung cancer. Albumin was often used by clinicians to assess a patient's nutritional status, low albumin levels indicated possible disease, including nephritis, hepatitis, cancer, and other infectious diseases (*Rozga, Piątek & Małkowski, 2013*).

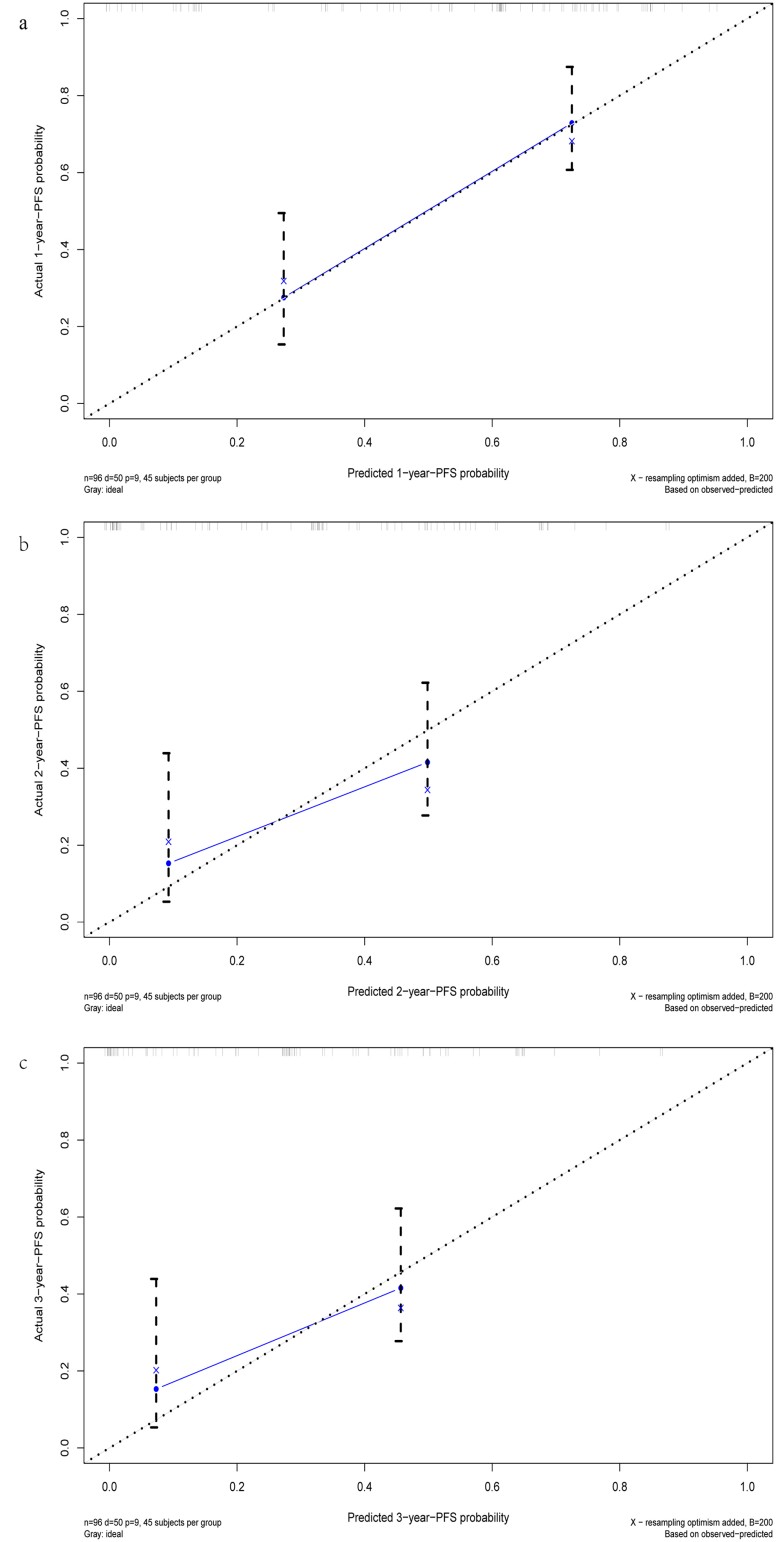

**Figure 3** **The calibration curves of the 1-, 2-, and 3-year PFS prediction nomogram. (A) The calibration curve for 1-year-PFS. (B) The calibration curve for 2-year-PFS. (C) The calibration curve for 3-year-PFS.**

In physiological state, albumin in bloodstream mainly played a regulatory role, assisting the body to transport various substances such as hormones, bilirubin, fatty acids and others to organs (*Don & Kaysen, 2004*). Systemic inflammation often led to hypoalbuminemia, and the invasion and metastasis of cancer were closely associated with the inflammatory state and immune status of the body. Hypoalbuminemia occurred frequently in cancer patients, mainly as a result of the body's consumption of excessive tumor load, and possibly due to low liver synthesis.

Monocytes were the largest blood cells in the blood. It was believed that monocyte was the predecessor of macrophage, which had obvious deformation movement and can engulf senescent cells and their fragments. Monocytes also participated in the immune response and could expose the antigen determinant to lymphocytes after phagocytosing antigen, then induced the lymphocyte specific immune response. Monocytes were also the main cellular defense system against intracellular pathogenic bacteria and parasites, and have the ability to recognize and kill tumor cells. Although some studies had suggested that monocytes were negative regulators of immune response, they can differentiate into macrophages, in the tumor microenvironment, M2 macrophages were the dominant tumor-associated macrophages, and M2 macrophages can secrete a large number of immunosuppressive factors, and T-lymphocyte responses were inhibited by these inhibitory factors and cells, thus unable to perform effectively (*Chen et al., 2021*; *Pan et al., 2020*). This conclusion was inconsistent with our results. However, peripheral blood monocytes was not completely consistent with those in tumor microenvironment, and once patients received immunotherapy, the inhibitory state of tumor microenvironment would change, which may induced the differentiation of monocytes into M1 macrophages, thus facilitating the antitumor effect of ICIs.

At present, immunotherapy had become one of the standard agents for lung cancer patients, and has been used in first-line, after first line, and neo-adjuvant therapy agents of lung cancer (*Jia et al., 2020*; *Liu et al., 2021*; *Geraci & Chablani, 2020*). However, no studies have compared the efficacy of immunotherapy in the first line with that in the back line, and our study explored this and proved that the first-line immunotherapy can improve the PFS, ORR and DCR of patients, which can be applied as one of biomarkers to predict the response to immunotherapy. Previous studies had suggested age as a prognostic marker for cancer patients, especially for breast cancer, the younger the patient, the more aggressive the tumor. Expert consensuses had recommended age <35 years as a discriminatory characteristic of increased risk and have applied age as one of factors determining therapy strategies (*Goldhirsch et al., 2007*; *Goldhirsch et al., 2001*). The body's immune system changes dynamically as we age. The effectiveness of immunotherapy was closely related to the state of the immune system, so it may also be age-related. One meta-analysis showed that OS benefit of immunotherapy was found for both younger (<65 years: HR, 0.77; 95% CI [0.71–0.83]) and older (≥65 years: HR, 0.78; 95% CI [0.72–0.84]) patients. No significant difference was found between two groups (*Yang et al., 2020*). Clinical studies have also manifested that older patients were more likely to benefit from immunotherapy (*Fehrenbacher et al., 2018*; *Fehrenbacher et al., 2016*). Basic researches have demonstrated that younger people and mice were susceptible to develop drug resistance to immune

checkpoint inhibitors, resulting in a suppressed tumor microenvironment. Older patients had a smaller proportion of Treg/CD8+T cells than younger patients (*Kugel et al., 2018*). Our study also demonstrated age as an indicator of immunotherapy outcomes, older patients were more likely to benefit from immunotherapy. Considering that age, LDH, line, ALB and MON were easy to obtain and had a high cost-effectiveness ratio in clinical practice, we combined age, ALB, LDH, line, and MON together to develop a prediction model, the reliability of the model was evaluated by internal validation and C-index calculation. The nomogram will help clinicians make treatment decisions and bring the best therapy strategy for patients with cost-effectiveness.

The research also has some limitations due to its retrospective and single-center nature. Larger sample sizes were needed to validate the predictive value of age, LDH, line, ALB and MON for immunotherapy. Our study did not analyze the relationship between baseline characteristics and patients' OS after immunotherapy. On the one hand, some patients were lost to follow-up, and on the other hand, some patients' family members were reluctant to tell patients the specific date of death during telephone follow-up, which led to the loss of most follow-up data. Our study may provide a basis for further exploration of this association. In addition, whether age, LDH, line, ALB and MON have predictive value in other treatment options for lung cancer patients also needs to be further explored, which can further determine the specific correlation of age, LDH, line, ALB and MON with immune response and inflammation.

## CONCLUSIONS

In summary, in this study of 216 patients with advanced lung cancer receiving immunotherapy, age, LDH, line, ALB and MON were significantly associated with PFS, response rate and cancer control. Older patients, higher MON, lower LDH, higher ALB and first line predicted better PFS and higher probability of response rate and tumor control. The nomogram has great predictive value for the efficacy of immunotherapy. All these indicators including age, LDH, line, ALB and MON were minimally invasive and easily accessible indicators, which can provide clinicians with timely and valuable references for selecting patients for immunotherapy.

## ACKNOWLEDGEMENTS

We thank the Tianjin Medical University Cancer Institute & Hospital for providing us with retrospective cases.

### Funding

This present research was funded by the National Natural Science Foundation of China (to Richeng Jiang) (No. 82172620). The funders had no role in study design, data collection and analysis, decision to publish, or preparation of the manuscript.

## Grant Disclosures

The following grant information was disclosed by the authors:
National Natural Science Foundation of China: 82172620.

## Competing Interests

The authors declared that they have no competing interests.

## Author Contributions

- Xiuqiong Chen conceived and designed the experiments, analyzed the data, authored or reviewed drafts of the article, and approved the final draft.
- Zhaona Li performed the experiments, prepared figures and/or tables, and approved the final draft.
- Jing Zhou performed the experiments, authored or reviewed drafts of the article, and approved the final draft.
- Qianhui Wei analyzed the data, authored or reviewed drafts of the article, and approved the final draft.
- Xinyue Wang performed the experiments, analyzed the data, prepared figures and/or tables, and approved the final draft.
- Richeng Jiang conceived and designed the experiments, analyzed the data, authored or reviewed drafts of the article, and approved the final draft.

## Data Availability

The raw measurements are available in the Supplemental Files.

## Supplemental Information

Supplemental information for this article can be found online at http://dx.doi.org/10.7717/peerj.14566#supplemental-information.

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
