# Peer review of "Identification of prognostic factors and nomogram model for patients with advanced lung cancer receiving immune checkpoint inhibitors"

_PeerJ, doi:10.7717/peerj.14566_

## Round 0.1 · original submission · Major Revisions

Dear Dr. Chen,

Thank you for submitting your manuscript, "Identification of prognostic factors and nomogram model for patients with advanced lung cancer receiving immune checkpoint inhibitors" to PeerJ. We have now sufficiently received reports from three reviewers. After careful consideration, we have decided to invite a major revision of the manuscript.

As you will see from the reports copied below, the reviewers raise important concerns regarding the rationale for some of the experiments, which includes investigating monocyte over lymphocyte, lack of PD-L1 expression, use of statistical methods, information related to drug types and blood analysis to identify HB, LDH and ALB levels. We find that these concerns limit the strength of the study, and therefore we ask you to address all of the reviewers' comments with additional work. Without substantial revisions, we will be unlikely to send the paper back for review.

Important:
If you feel that you are able to comprehensively address the reviewers’ concerns, please provide a point-by-point response to these comments along with your revision. Please show all changes in the manuscript text file with track changes or color highlighting. If you are unable to address specific reviewer requests or find any points invalid, please explain why in the point-by-point response.

Best regards,

Abhishek Tyagi, PhD
Academic Editor,
PeerJ

Reviewer 1 ·

Basic reporting

No comments

Experimental design

This study aims to analyze predictors of response to immunotherapy, and in many cases, cytotoxic agents were combined with ICI. It is inappropriate as an analysis of predictors of response to immunotherapy.

PD-L1 expression in tumors is accepted as an established predictor of response to immunotherapy. Because of the possibility of potential bias, PD-L1 expression should be added to the analysis.

The authors adopted monocyte counts among the leukocyte fractions, however, lymphocyte counts also showed significant differences in univariate analysis. As the authors mentioned, lymphocyte counts have been widely reported as a prognostic factor in past studies. Please provide the rationale as to why monocytes were preferentially selected.

Regarding age, many reports suggest that immunotherapy is less effective in the elderly. Please discuss in your discussion why the elderly had a better prognosis in this study.

Validity of the findings

The prognostic factors identified in this study have been reported in previous studies. It is difficult to validate this study due to the presence of potential bias using data from a limited number of patients.

Reviewer 2 ·

Basic reporting

no comment

Experimental design

This study is a well-structured retrospective analysis,the authors constructed prognosis model based on inflammation related blood indicators and clinical signatures. In general, the sample size of the research is sufficient and the statistical analysis process is reasonable.

Validity of the findings

The innovation of this study lies in the lack of effective predictors of immunotherapy at present and the rarity of immunotherapy cohorts with large samples.

Additional comments

1. Now that the authors have data on the efficacy of remission after immunotherapy, why not use logistic regression to establish diagnostic models to directly predict the efficacy of immunotherapy? Because the diagnostic model can at least tell if the patient is more likely to respond or not respond to immunotherapy. If a patient wants to use this prognostic model to know whether he or she should receive immunotherapy, what are the criteria for such differentiation? Please explain it.
2. Please supplement this immunotherapy cohort with information on drug types.

Reviewer 3 ·

Basic reporting

Some patients with lung cancer can benefit from immunotherapy, but the biomarkers that predict immunotherapy response were not well defined. Baseline characteristic of
patients may be one of the most convenient and effective markers. In the manuscript “Identification of prognostic factors and nomogram model for patients with advanced lung cancer receiving immune checkpoint inhibitors”, authors found that Age, ALB, MON, LDH, line can be used as reliable predictive biomarkers for PFS, response rate and cancer control in patients with lung cancer receiving immunotherapy. Besides, the nomogram based on age, ALB, MON, LDH, line was of great significance for predicting 1-year-PFS, 2-year-PFS and 3-year-PFS in patients with advanced lung cancer treated with immunotherapy.

Experimental design

This paper is planned correctly and well documented, while the methods used in the work are modern and correct.

Validity of the findings

Moreover, work is written in good English with well presentation and layout, and the figures and tables were vivid. In addition, the conclusions are correct and cautious.

Additional comments

However, there are several minor issues that if addressed would dramatically improve the manuscript.
1 There were similar reports (Ann Transl Med. 2020 Apr;8(7):470; Transl Lung Cancer Res. 2021 Dec;10(12):4511-4525) about the constructed nomogram based on peripheral blood lymphocyte subsets to assess the prognosis of non-small cell lung cancer patients treated with immune checkpoint inhibitors in PubMed. Please cite and compare in Introduction part.
2 What were the roles of WBC, LYM, MON, AEC, HB, LDH and ALB in the treatment with immune checkpoint inhibitors (ICIs), respectively? Please state in the introduction.
3 Whether there were different efficacies in different immune checkpoint inhibitors?
4 How to analyze blood samples to detect WBC, LYM, MON, AEC subsets and identify HB, LDH and ALB levels?
5 What were the effects of immune checkpoint inhibitors on WBC, LYM, MON, AEC subsets? Please supplement in the discussion.
6 Some language mistakes and format errors should be revised.

Annotated reviews are not available for download in order to protect the identity of reviewers who chose to remain anonymous.

---

## Round 0.2 · accepted · Accept

Dear Dr. Chen,

We are delighted to accept your manuscript, entitled "Identification of prognostic factors and a nomogram model for patients with advanced lung cancer receiving immune checkpoint inhibitors," for publication in PeerJ. Thank you for choosing to publish your interesting work with us.


With kind regards,
Abhishek Tyagi
Academic Editor, PeerJ

Reviewer 2 ·

Basic reporting

no comments

Experimental design

no comments

Validity of the findings

no comments

Additional comments

After modification, the author basically answers the main questions and is supplemented accordingly in the article.This paper has clear logic, no obvious errors, and has certain innovative and clinical significance. I think the mannuscrips could be accepted.